# Spotting the Targets of the Apospory Controller *TGS1* in *Paspalum notatum*

**DOI:** 10.3390/plants11151929

**Published:** 2022-07-26

**Authors:** Carolina Marta Colono, Maricel Podio, Lorena Adelina Siena, Juan Pablo A. Ortiz, Olivier Leblanc, Silvina Claudia Pessino

**Affiliations:** 1Laboratorio de Biología Molecular, Instituto de Investigaciones en Ciencias Agrarias de Rosario (IICAR) CONICET-UNR, Facultad de Ciencias Agrarias, Campo Experimental Villarino, Universidad Nacional de Rosario, Zavalla S2125ZAA, Santa Fe, Argentina; colono@iicar-conicet.gob.ar (C.M.C.); podio@iicar-conicet-gob.ar (M.P.); siena@iicar-conicet.gob.ar (L.A.S.); ortiz@iicar-conicet.gob.ar (J.P.A.O.); 2DIADE, University of Montpellier, IRD, CIRAD, 34394 Montpellier, France; olivier.leblanc@ird.fr

**Keywords:** apomixis, apospory, *LHC Ib-21*, miR2275, *QGJ*, *TGS1*

## Abstract

Sexuality and apomixis are interconnected plant reproductive routes possibly behaving as polyphenic traits under the influence of the environment. In the subtropical grass *Paspalum notatum*, one of the controllers of apospory, a main component of gametophytic apomixis reproduction, is *TRIMETHYLGUANOSINE SYNTHASE 1* (*TGS1*), a multifunctional gene previously associated with RNA cleavage regulation (including mRNA splicing as well as rRNA and miRNA processing), transcriptional modulation and the establishment of heterochromatin. In particular, the downregulation of *TGS1* induces a sexuality decline and the emergence of aposporous-like embryo sacs. The present work was aimed at identifying *TGS1* target RNAs expressed during reproductive development of *Paspalum notatum*. First, we mined available RNA databases originated from spikelets of sexual and apomictic plants, which naturally display a contrasting *TGS1* representation, to identify differentially expressed mRNA splice variants and miRNAs. Then, the role of *TGS1* in the generation of these particular molecules was investigated in antisense *tgs1* sexual lines. We found that *CHLOROPHYLL A-B BINDING PROTEIN 1B-21* (*LHC Ib-21*, a component of the chloroplast light harvesting complex), *QUI-GON JINN* (*QGJ*, encoding a MAP3K previously associated with apomixis) and miR2275 (a meiotic 24-nt phasi-RNAs producer) are directly or indirectly targeted by *TGS1*. Our results point to a coordinated control exercised by signal transduction and siRNA machineries to induce the transition from sexuality to apomixis.

## 1. Introduction

Apomixis is a revolutionary trait in terms of increasing food production on a sustainable basis [1]. This peculiar mode of plant reproduction, described in at least 293 angiosperm genera [2], consists of the spontaneous formation of maternal seeds in absence of meiotic recombination and fertilization [3]. The joint use of sexuality and apomixis in plant breeding programs allows for a permanent fixation of heterotic genotypes, thus accelerating work outlines and reducing hybrid seed costs [4]. Apomixis-based programs aimed at developing new adapted varieties basically consist of crossing sexual mother plants and apomictic pollen donors, followed by selecting superior F_1_ hybrids with asexual reproduction capacity. In theory, a single cross involving any genotype, even a highly heterozygous one, might suffice to produce improved varieties capable of cloning themselves by seeds [4]. Such relatively simple breeding schemes are currently operative in apomictic forage grasses of the *Paspalum* and *Brachiaria* genera, but still remain unfeasible in sexual major crops [5,6,7]. However, widely cultivated species such as sorghum, sunflower or cassava show reproductive anomalies mimicking particular apomixis steps [8,9,10,11]; this allows for raising the hypothesis that, at least for some species/genotypes, the induction of full apomixis capacity through the establishment of particular endogenous/exogenous conditions remains a plausible outcome. Furthermore, several major sexual crops such as rice, wheat, pearl millet or maize, have apomictic relatives from which the trait could potentially be introgressed by crossing [12,13,14,15]. Finally, the notion of possible expression in major crops was reinforced by recent viewpoints presenting apomixis and sexuality as ancient phenisms [16]. In any case, detailed information on the molecular pathways controlling the trait is imperative to ensure effective and responsible use in agriculture. Risks related with pollen transfer and introgression into sexual crops, escapes into natural populations, and potential invasiveness of apomictic plants may need the use of specific phenotypes (i.e., pollen-sterile or cleistogamous variants, polyploids), cautious management and wide-ranging consideration of ecological impact [17].

In sexual plants, female reproductive development starts with the differentiation of a single megaspore mother cell (MMC) in the ovule primordium. This MMC divides by meiosis to form four haploid megaspores from which, most frequently, only the chalazal one survives and progresses to form an octa-nucleate syncytium after three rounds of mitoses [18]. Further development consists of a differentiation/cellularization process giving rise to a haploid organism (the megagametophyte), carrying seven differentiated cells at defined positions: an egg and two synergids (i.e., the egg apparatus) close to the micropyle, one binucleated central cell and three chalazal antipodal cells. To start embryogenesis, the reduced egg cell and the central cell are fertilized by pollen generative nuclei, to form the zygote and the first endosperm nucleus, respectively, a process in which the maternal:paternal genomic contribution is strictly controlled [18]. Apomixis short-circuits the sexual pathway described above, possibly as a consequence of a chronological and spatial molecular deregulation induced by a few leader genes, getting away with the formation of maternal seeds. Gametophytic apomicts avoid meiosis and form nonreduced megagametophytes from a 2n cell, either the MMC itself (diplospory) or nucellar companion cells (apospory). Egg cells included in these non-reduced megagametophytes are capable to generate embryos by parthenogenesis, while the endosperm can develop autonomously of after pseudogamy (i.e., fertilization of the central cell only) [19].

*Paspalum notatum*, a perennial subtropical grass native to South America, has served as one of the traditional models for the study of the molecular basis of aposporous apomixis in grasses [20,21]. Moreover, it is recognized as a valuable forage grass, which also provides benefits in terms of soil conservation [20,21]. The species includes sexual self-incompatible diploids and aposporous self-fertile pseudogamous tetraploids, which can grow in sympatry or isolation [22,23,24]. Population studies revealed that sexual diploids can occasionally form new apomictic polyploids via hybridization of non-reduced gametes. Therefore, the genetic variability present in sexual diploid populations can be pumped up to apomictic polyploid ones [22,23,24,25]. Sexual tetraploid plants were never found in nature, but they were experimentally obtained [26] and then used to produce segregating populations in which a single genomic region governing apomixis (i.e., the Apomixis Controlling Locus or ACL) was identified, mapped and partially characterized [27,28,29,30]. The *P. notatum* ACL consists of a chromosomal region of about 36 Mbp, syntenic to specific segments of rice chromosomes 2 and 12 [27,28,31]. It shows a severe restriction of recombination [27,28], as well as heterochromatin features [30]. Comparative PCR-based and NGS analyses involving florets from sexual and apomictic individuals allowed the identification of numerous differentially expressed transcripts (DETs) originated from long and short RNA components [32,33,34,35,36], some of which map into the ACL region [21].

One of the apo/sex DETs emerging from the above-mentioned analyses is *TRIMETHYLGUANOSINE SYNTHASE 1* (*TGS1*) [33]. Quantitative PCR and in situ hybridization involving genotypes with different degrees of facultative apomixis revealed the representation of this transcript was positively correlated with the level of sexual reproduction [37]. At premeiosis/meiosis, *TGS1* was found expressed in ovules and anther tapetum of sexual plants, reaching a maximum representation at anthesis, while its concentration remained relatively low in apomictic individuals [37]. Moreover, a downregulation of the *TGS1* in sexual plants through antisense technology caused a decrease in the proportion of ovules carrying meiotic embryo sacs and a concurrent emergence of supernumerary megagametophytes resembling aposporous embryo sacs [38]. No signal of parthenogenesis was detected in *tgs1* antisense lines and seeds showed a decrease in the germination capacity, which suggests a possible additional role during endosperm development and/or embryogenesis [38]. 

In other eukaryote systems (yeasts, mammals, drosophila), *TGS1* was initially cloned and characterized as an RNA binding protein with a methyltransferase domain, which enhances the PRIP (PPAR-interacting protein) nuclear receptor coactivator function [39]. It can act as a transcriptional coactivator bridge that, in presence of a ligand-bound PPAR, connects the CBP/p300-anchored coactivator complex (with histone acetyltransferase activity) with the PBP-anchored TRAP/DRIP/ARC core, facilitating the recruitment of general transcription factors (GTFs) and RNA polymerase II holoenzymes, to initiate transcription of specific target genes [40,41]. Moreover, at least in yeasts and animals, the *TGS1* methyltransferase domain is involved in trimethylation of the capping of different RNA (snRNAs, snoRNAs, telomerase RNA, pri-pre miRNAs), with direct impact in RNA processing [42,43,44]. This gene is essential to meiosis in *Saccharomyces* [45] as well as to development in *Drosophila* [46,47] and mice [48]. Recently, *TGS1* was shown to control SWI6/HP1-independent siRNA production and the establishment of heterochromatin in fission yeast [49].

To initiate the identification of molecular pathways modulated by *TGS1* in *P. notatum* ovules, we decided to evaluate a group of RNA molecules showing differential processing in plants with attenuated *TGS1* expression. First, we compared sexual and apomictic floral transcriptomes in search for differentially represented mRNA splice variants and miRNAs, taking advantage of the contrasting *TGS1* activity previously described for these reproductive biotypes [37]. Then, we carried out intron-specific (for mRNAs) or stem-loop (for miRNAs) qPCR analysis in sexual *tgs1* antisense lines as well as in apomictic and sexual controls, to confirm a causal association between the *TGS1* knock-down and the emergence of particular splice variants/miRNAs. Using this procedure, we identified three transcripts whose processing or expression is altered in *tgs1* antisense lines, i.e., *LHC Ib-21* (encoding CHLOROPHYLL A-B BINDING PROTEIN 1B-21, a protein with reported RNA binding activity) [50], *QGJ* (encoding a MAP3K previously associated with aposporous development) [51] and miR2275 (shown to produce PHAS siRNAs essential to meiosis in rice) [52]. Our results provide information on *TGS1* functional targets, open the way to further analysis of apomixis candidates, and contribute to assemble the molecular puzzle redirecting plant seed reproduction from sexuality to asexuality.

## 2. Results

### 2.1. The Emergence of a Particular LHC Ib-21 Splice Variant Is Influenced by TGS1

Considering that *TGS1* is required for specific RNA processing/cleavage in several organisms [42,43], we started by comparing the representation of particular mRNA splice variants in florets of apomictic and sexual plants, in which *TGS1* was shown to be contrastingly expressed (i.e., overexpressed in ovules and anther tapetum of sexual plants) [37]. We first surveyed a list of DETs expressed in flowers of apomictic and sexual plants, which had been identified via RNAseq [34]. In particular, we focused on the 316 DETs with the lowest false discovery rates (FDRs < 6.74 × 10^−10^) from the 3732 ones differentially represented between apomictic and sexual libraries at *p*-values ≤ 0.01 and logFC ≥ |3| [34]. The full mRNA sequences of these 316 transcripts, which had been assembled from reads originated from both sexual and apomictic libraries (Global Assembly), were used to recover all homologs from the apomictic and the sexual floral transcriptomes. Alignments of sexual and apomictic isoforms and BLAST surveys on the NCBI green plants nucleotide databases revealed that 20 DETs represented possible splice variants (Appendix A). Six of them were chosen for subsequent analysis, based on the following criteria: (1) putative differential processing occurring within internal regions of the transcript (to discard incomplete sequencing); (2) presumed alternative splicing sectors representing characterized introns in other species and (3) presence of the intron typical context GU-AG. A graphic displaying the structure of the six selected transcripts’ putative splice variants from apomictic and sexual libraries are provided in Figure 1. To construct Figure 1, the variants carrying introns were used as query, to reveal the absence of this part of the sequence in the rest of the variants. Annotations and relevant statistical parameters are listed in Table 1.

Next, we designed PCR primer pairs to amplify internal regions of those splice variants that carried introns. One primer was invariably located inside the target intron and the other one on the adjacent exon, to produce PCR amplicons specific for the non-processed variant (+intron) (Figure 1, Appendix A. Note that the processed variants (−intron) cannot be amplified without coamplifying the non-processed ones (+intron) (i.e., primers located on exons surrounding the missing intron amplify products of different size from both variants). Therefore, successful qPCR quantification can be achieved only for the non-processed variant (+intron). For i10779, three primer pairs were designed, corresponding to different exon-introns boundaries.

Initially, non-quantitative PCR reactions were used to check the number of bands produced by each primer combination in cDNA from florets of an obligate apomictic (Q4117) and a full sexual (C4−4x) genotype (Figure 2A). Successful fragment amplification in both plants was obtained for i10779, i23387, i11548, i22343, and i24572, while no band neither from the apomictic nor the sexual genotypes were obtained for i22630, even after applying different cycling conditions (Figure 2A). While i10779 and i23387 homologs amplified single bands, those corresponding to i11548, i22343 and i24572 amplified more than one isoform, possibly due to the occurrence of overlapped splice variants or unspecific annealing. Therefore, they were discarded for further qPCR analysis, which requires the generation of a single product.

Real-time qPCR assays were used to amplify i10779 and i23387 on floral cDNA samples from three obligate apomictic and three fully sexual genotypes, using the same primer pairs tested in the non-quantitative PCR analyses (Figure 2B, Appendix A). While i10779 amplification levels did not significantly differ between apomictic and sexual samples (not shown), i23387 displayed higher relative expression levels in sexual genotypes in comparison with apomictic ones (Figure 2B, Appendix A).

Since the i23387 processed (−intron) variant cannot not be specifically amplified in qPCR experiments, we decided to examine its expression in recently delivered Illumina RNAseq libraries originated from florets of sexual and apomictic genotypes at different developmental steps (premeiosis, meiosis, postmeiosis, anthesis) [36]. Blast analysis revealed it corresponded to transcript TRpn_180064 [36] (ID: 100%; E-value: 0.0) and was overexpressed in apomictic libraries respect to sexual ones at anthesis (logFC: 17516; *p*-adjusted value: 0.0091) [36] (Figure 3A). Moreover, the number of reads detected for this processed (−intron) variant was well above of that of the non-processed (+intron) one (transcript TRpn_99393) (ID: 99.85%; E-value: 0.0), which was represented at low levels in the libraries and for which no significant differential expression could be detected, due to low count (Figure 3A). In summary, our results indicate in flowers of apomictic *P. notatum* plants there is a drastic upregulation of the processed (−intron) form of i23387, while the non-processed (+intron) form remains at low levels in both genotypes but, at least according to the more sensitive qPCR results, it is overexpressed in sexual ones.

To investigate a possible causal association between the differential *TGS1* representation detected in apomictic and sexual plants [37] and the occurrence of particular splice variants of the i23387 homologs, we examined two independent *tgs1* lines (*tgs1-1*: E2.9 and *tgs1-2*: E2.13) previously generated by transformation of a sexual *P. notatum* genotype with a construction carrying an antisense copy of *TGS1* under the rice *act1* promoter [38]. These antisense lines display a constitutively attenuated *TGS1* expression, form putative apospory initials, supernumerary gametophytes with a morphology resembling AESs as well as abundant leaf trichomes [38]. qPCR analyses revealed a significantly higher expression of the non-processed (+intron) i23387 isoform in florets of the sexual control with respect to the two *tgs1* antisense lines (Figure 2C, Appendix A). These results suggest that *TGS1* participates of the i23387 transcript processing. Moreover, they are in coincidence with the higher relative expression levels of non-processed i23387 form detected in sexual flowers (where *TGS1* is highly expressed) in comparison with apomictic flowers (where *TGS1* is naturally downregulated). In agreement with the ontology analysis made by Ortiz et al. (2017) [34], our NCBI BLASTX surveys confirmed significant homology (E-value: 3 × 10^−121^; % ID: 84.21%) between i23387 and *CHLOROPHYLL A-B BINDING PROTEIN 1B-21 (LHC Ib-21)* of *Setaria italica* (XP_004965344) (alternative names: *LHCI TYPE I CAB-1B-21*; *LHCI-730 CHLOROPHYLL A/B BINDING PROTEIN*; *LIGHT-HARVESTING COMPLEX I 21 KDA PROTEIN*). Even when the detection of a photosynthesis related transcript was not expected, a possible relation between *LHC Ib-21* and development is commented in the discussion. 

### 2.2. The Expression of the MAP3K QUI-GON JINN (QGJ) Is Influenced by TGS1 

As an independent strategy to identify additional *TGS1* targets, we focused on the mitogen activated protein kinase kinase kinase (MAP3K) QUI-GON JINN (QGJ), which is necessary for the formation of AESs in *Paspalum* [51]. Since attenuation of *TGS1* leads to the development of AES-like gametophytes in the same species [38], we decided to investigate a possible functional link between both genes. *QGJ* is represented by two splice variants in apomictic and sexual plants [51]. We used non-specific primers (Appendix A) to measure: (i) the relative *TGS1* expression levels in two constitutive *qgj* RNAi lines (*qgj-1*, *qgj-2*) generated by transforming an apomictic genotype with a hairpin *QGJ* construction [51]; and (ii) the relative *QGJ* expression levels in one of the constitutive *tgs1* (*tgs1-1*) antisense line used in the former section [38]. No differential *TGS1* expression was detected in the *qgj* defective lines with respect to an apomictic control at anthesis (when expression of *TGS1* peaks [37]; the apomictic control used here was the same genotype originating the *qgj* defective line by biolistic transformation, i.e., Q4117) (Figure 4A, Appendix A). Contrarily, a statistically significant downregulation of *QGJ* expression was detected in both the *tgs1* defective line and the apomictic control with respect to the sexual control at meiosis (when the highest *QGJ* expression is detected [51]; the sexual control was C4-4x; the apomictic control was Q4117) (Figure 4B, Appendix A). The latest result indicates that a downregulation of *TGS1* have the same impact in the total floral concentration of *QGJ* at meiosis as observed in apomictic plants (a decrease, possibly due to downregulation in pollen mother cells, according to Mancini et al. 2018) [51]. 

Next, we investigated the previously-reported occurrence of two *QGJ* splice variants. First, we used a primer pair that specifically identifies the *QGJ* non-processed (+intron) splice variant in qPCR experiments (Appendix A). No significant variation was observed for this particular isoform (Figure 4C,D, Appendix A), which is in agreement with results presented by Mancini et al. (2018) [51], who detected no differences in levels of the non-processed variant in sexual and apomictic plants. Moreover, we checked the Illumina libraries published by Podio et al. (2021) [36] and found the non-processed (+intron) splice variant (TRpn_120075) (ID 96.96%; E-value: 0.0) equally expressed in sexual and apomictic samples at all developmental stages, while the processed (−intron) variant (TRpn_64321) (ID 99.93%; E-value: 0.0) resulted significantly upregulated in sexual plants at premeiosis, and remained upregulated at other developmental stages but at statistically non-significant levels (Figure 3B). Altogether, these data indicate that, at premeiosis/meiosis, a downregulation of *TGS1* causes an alteration in the expression level of the processed (−intron) *QGJ* isoform but not in the non-processed (+intron) one. 

### 2.3. In Situ Analysis of QGJ Expression in tgs1 Antisense Lines

As we detected a change in the quantitative expression level of *QGJ* in florets of antisense *tgs1* lines, in situ experiments were designed to test if there were modifications in the spatial distribution pattern of *QGJ* in the ovule, as it was observed in aposporous plants with respect to sexual ones (Figure 5). At premeiosis, in both *tgs1* antisense lines and wild type sexual plants, the *QGJ* signal was detected across the whole nucellus and within the MMC, but while in sexual genotypes the signal was generally robust, in *tgs1* lines was rather fainter (Figure 5A,D). During meiosis, in the control sexual plants the *QGJ* expression shifted to the micropylar region and was also observed in the funiculus, but had lower levels in the chalaza (Figure 5B); the degenerating megaspores showed moderate to intense signal, but the functional one (at a more chalazal position) had no signal. On the contrary, during meiosis, in *tgs1* antisense lines the signal moved to the chalaza instead of the micropyle, and was also observed in the funiculus as well as in all meiotic products (Figure 5E). Regarding male development, in control sexual genotypes a consistent signal was observed within pollen mother cells (PMCs) but not in the tapetum (Figure 5C). Nevertheless, in anthers of *tgs1* genotypes there was moderate signal in the tapetum, while PMCs showed heterogeneous staining (i.e., some of them displayed no signal, while others did) (Figure 5F). The sense probe showed undetectable hybridization signals (Figure 5G–J). In general, results obtained here for the *tgs1* antisense lines are analogous to those presented by Mancini et al. (2018) [51] for florets of apomictic *P. notatum* and *Brachiaria brizantha* plants. Additional images of the in situ experiments were provided in Appendix A. This evidence strongly suggests that the silencing of *TGS1* causes a shift of *QGJ* expression from the micropyle to the chalaza in premeiotic/meiotic ovules, and from pollen mother cells to the tapetum in premeiotic/meiotic anthers, which means that *QGJ* expression moves from a typical sexual expression pattern to an apomictic-like pattern.

### 2.4. Identification of miRNA Variants Associated with TGS1 Activity

In human fibroblasts, *TGS1* mediates the cleavage of a group of pri-pre-miRNA precursors in order to produce mature miRNAs associated with EXPORTIN 1 [44]. We could not identify the orthologs to such group of miRNAs in *Paspalum*, possibly due to low conservation. However, since *TGS1* is naturally upregulated in spikelets of sexual plants, we decided to search miRNAs which were overrepresented in floral transcriptome libraries of sexual plants with respect to apomictic ones. To do so, we exploited the *P. notatum* floral small RNA libraries of apomictic and sexual individuals generated by Ortiz et al. (2019) [35]. Based on the quantitative analysis of miRNA representation, only one miRNA (miR2275) was detected upregulated in spikelets of sexual plants in comparison with apomictic ones. miR2275 showed a normalized counting of 23 reads in the apo libraries and 48 in the sexual ones [35]. Meanwhile, in long-read Roche 454/FLX + libraries involving the same floral samples, the miR2275 precursor (Cluster_47025, sex_isotig38116) was detected twice in the sexual sample but was absent from the apo one [34].

In order to assess the miR2275 in vivo representation within in the *P. notatum* floral system, a stem-loop PCR assay was designed to amplify the mature miRNA in sexual control plants and a *tgs1* defective line (Appendix A). At meiosis, miR2275 resulted significantly downregulated in the *tgs1* line, reaching levels closer to those detected in apomictic plants (Figure 6A, Appendix A), yet still significantly higher. As miR2275 was reported as a potential *AGO1* repressor in *Paspalum* [35], we decided to check the levels of miR168 (a characterized *AGO1* repressor) in ovules of sexual/apomictic controls and *tgs1* defective lines. Interestingly, miRNA168 was also found differentially expressed in sexual and aposporous plants, with overexpression in apomictic ones [35]. However, our stem-loop PCR analysis showed similar levels of miR168 representation in *tgs1* defective and sexual lines, which were, as expected, lower that those observed in apomictic lines (Figure 6B, Appendix A).

These results show that two potential *AGO1* controllers (miR2275 and miR168) are differentially expressed in apomictic and sexual *P. notatum* ovules at premeiosis (upregulated in sexual and apomictic plants, respectively), but only one of them (miR2275) seems to be modulated by *TGS1*. Then we checked the *AGO1* expression in the available *P. notatum* floral transcriptome libraries of sexual and apomictic plants [36]. We found 21 detectable AGO1-like transcripts, from which 8 are expressed at considerable levels. The most expressed transcript (TRpn_87312) is significantly upregulated in sexual plants at anthesis [36]. The rest of the transcripts are significantly upregulated in apomictic plants at all stages (TRpn_171351, TRpn_104454, TRpn_153031, TRpn_176544), at premeiosis, meiosis and anthesis (TRpn_86127), at postmeiosis and anthesis (TRpn_109434) or only at meiosis (TRpn_103957) [36].

## 3. Discussion

In the last few years, an unprecedented amount of data originated from genome and transcriptome sequencing projects have flooded the apomixis field. Hundreds of candidate genes showing differential expression in sexual and apomictic plants were identified at different reproductive developmental stages [53]. However, little is known on the functional role of these candidates, the operative interactions among them or the identity of the molecules controlling the coexistence of a subtle balance between both reproductive types in natural populations. The identification of molecular markers co-segregating with apomixis, the generation of artificial sexual polyploids after colchicine duplication, the construction of transcriptome and genomic databases and the establishment of biolistic transformation platforms represent good perspectives for selecting traits of interest in natural apomictic species such as *P. notatum* [21]. In fact, heterosis for forage yield and cold tolerance has been repeatedly reported in this species, and an upright and fast-growing apomictic *P. notatum* hybrid was recently released as a forage cultivar [21]. Instead, the harnessing of apomixis in major crops such as rice and maize requires a much more detailed knowledge of the molecular mechanisms controlling the balance between apomixis and sexuality. The establishment of such programs requires sophisticated molecular tools and might bring new ecological challenges related with spreading the trait via pollen and seeds (e.g., uniparental reproduction, unidirectional gene transfer) [17]. Therefore, before any attempt of using the trait, we should expand our information on the molecular, functional and organizational mechanisms operative in natural apomictic plant populations, to improve our capacity to evaluate and avoid any damage. 

Natural apomicts display a wide range of developmental approaches to balance their competence for both genetic variation and cloning. Both features can coexist in the same species (i.e., confined at different ploidy levels), within the same plant (i.e., in facultative apomicts) and even the same ovule (i.e., occurrence of polyembryony of sexual and apomictic origin). Moreover, the proportion of offspring formed by each reproductive mode can be influenced by the environment [17]. In any case, there seem to be regulators operating to favor apomixis or sexual reproduction in different contexts (variable ploidy, diverse genetic backgrounds, particular environmental conditions), which could be harnessed to abolish the ecological impact of the trait ensuring a safe use in agriculture. Understanding the functional interactive dynamics of both reproductive modes at the molecular level will be crucial to predict how an apomictic crop may behave in natural fields and visualize potential ecological threats.

In connection with this, only a few sexuality/agamospermy switch regulators have thus far been reported in natural apomictic species [53]. Among them, *TGS1* is both a repressor of the formation of supernumerary AESs-like gametophytes from the nucellus of sexual individuals and a promoter of the meiosis capacity [38]. Once described as a gene with a physiological function (i.e., chilling tolerance) [54], it was also found differentially expressed in reproductive organs of sexual and apomictic *Paspalum* plants [37] and its attenuation in sexual individuals caused a reduction in the meiosis capacity and the emergence of AES-like gametophytes, pointing to an parallel function as a major developmental modulator of the apomixis-sexuality switch [38]. 

The availability of sexual *P. notatum* tetraploid plants with impaired *TGS1* activity [38] enabled studies aimed at identifying its target transcripts in *Paspalum notatum* ovules, in order to investigate the possible molecular pathways controlling the apomictic-sexuality connection. The first candidate we identified (*CHLOROPHYLL A-B BINDING PROTEIN 1B-21*) shows upregulation of a particular splice variant in sexual plants with respect to apomictic ones and *tgs1* lines, which might be related with the *TGS1* function as a major splicing controller as reported in other eukaryotic systems [42,43]. The modulation of a member of the *CHLOROPHYLL A-B BINDING PROTEIN 1B-21* in heterotrophic reproductive organs was an unanticipated result, since we were expecting candidates with a female reproductive function. However, light-induced phenotypes had already been described in antisense *tgs1* plants (i.e., the development of abundant leaf trichomes under a light regime) [38]. Besides, recently several components of the photosynthesis machinery were included within a group of RNA binding proteins (RBPs) with a key role in RNA metabolism [50]. Moonlighting functions have been hypothesized for these proteins, even when further analysis is required to understand the crosstalk between photosynthesis and RNA metabolism [50]. About 8% of the leaf RBPs described by Bach-Pages et al. (2020) [50] have annotations related to photosynthesis or photosystems, and many photosynthesis-related domains are enriched in the leaf RBPome, including the CHLOROPHYLL A-B BINDING domain. Moreover, 33 photosynthesis-related leaf RBPs have been independently shown to associate with RNA in other *Arabidopsis* tissues [55,56] and some photosynthesis components, such as the large subunit of rubisco (LSU) and cytochrome f, are known to bind RNA [57,58,59,60]. In future work, the link among the *TGS1* capacity to repress the formation of trichomes [38], the *TGS1*-mediated regulation of *LHC Ib-21* splice variants shown here and the capacity of *LHC Ib-21* to bind RNA [50] should be investigated in the context of the previously reported influence of the chloroplast on alternative splicing [61] and the well-known relation between trichome formation and light incidence.

We also investigated the functional link between *TGS1* and the transcript mitogen-activated 3-kinase *QGJ*, which was previously described as differentially represented in reproductive organs of sexual and aposporous plants and to be essential for AESs formation [51]. Natural apomictic plants show a downregulation of *TGS1* in the ovule chalaza and the anther tapetum [37], concurrent with a spatial deregulation of the *QGJ* representation (upregulation in chalaza, proximal nucellus and anther tapetum; downregulation in pollen mother cells and the cell layer surrounding the MMC) [51]. In qPCR analysis involving spikelets (somatic tissues, ovules and anthers), apomictic plants display lower quantitative *QGJ* levels with respect to sexual plants, since anthers represent most of the floret tissue [51]. Two alternative *QGJ* splice variants are expressed in *Paspalum* florets, one of them including an intron that introduces several additional aminoacids and shifts the open reading frame. However, no differential apo/sex representation was detected for the non-processed variant, the only one that can be evaluated in qPCR assays [34,51]. In our experiments, antisense *tgs1* lines showed a *QGJ* expression profile analogous to that of apomictic plants, suggesting that the attenuation of *TGS1* might be causing the deregulation of the *QGJ* expression detected during aposporous apomixis. While constitutive repression of *TGS1* in sexual plants induces the formation of AES-like gametophytes from somatic cells of the nucellus and/or the chalaza [38], down-regulation of *QGJ* in the ovule of aposporous plants impairs the development of AES [51], suggesting that both genes belong to antagonist pathways. Thus, the modulation exercised on *QGJ* could partially imply the activity of a non-cell autonomous mechanism, that, as we will discuss in the next paragraph, might be mediated by epigenetic mechanisms.

In human fibroblasts, *TGS1* was shown to control the processing of a group of miRNAs associated with the Exportin 1 function [44]. Given that the low degree of conservation among plants and human miRNAs prevents direct identification of homologs via sequence similarity analysis, and considering that sexual *P. notatum* plants have upregulated *TGS1* activity, we decided to examine the influence of *TGS1* expression on miRNAs overexpressed in sexual plants. Using previously available sRNA databases, we detected a single miRNA significantly upregulated in spikelets of sexual plants: miR2275 [35]. Stem-loop qPCR analysis revealed that this molecule was downregulated in *tgs1* lines at meiosis, reaching similar levels to those detected in apomictic plants. Interestingly, in rice and maize, miR2275 was shown to trigger a 24-nt phasiRNA pathway mediated by a specific DCL protein (DCL 5), which is essential to meiosis [52,62]. Recently, Xia et al. (2019) [63] reported that the miR2275/24-nt phasiRNA pathway is widely present in eudicots plants, but absent in well-characterized species of the *Fabaceae* and *Solanaceae*, as well as in the model plant *Arabidopsis*. In rice anthers, miR2275 is formed in the tapetum cell layer and trigger a 24-nt phasiRNAs pathway necessary for meiosis in pollen mother cells [52]. Migration of the 24-nt phasiRNAs from the site of origin (the tapetum) to pollen mother cells was proposed as the only possible regulatory mechanism to promote meiosis [62]. As commented above, in *Paspalum*, *TGS1* is expressed in the tapetum of sexual plants (a location coincident with miR2275 synthesis site in rice), as well as in the ovule nucellus/chalaza [37]. A decrease in *TGS1* expression is detected in apomictic plants at both locations [37], concurrently with a general decrease in the miRNA2275 levels [35], the formation of extra megagametophytes [38] and a reduction in pollen viability [27,64]. Moreover, *tgs1* antisense lines show lower miRNA2275 levels (results reported here) and emergence of AES-like structures [38]. An interesting point is that the *P. notatum* predicted a target of miR2275 is *AGO1* [35], a transcript previously implicated in meiotic development in yeasts [65]. *AGO1* was also found overexpressed in egg cells [66] and differentially expressed in male and hermaphrodite gametophytes of the homosporous fern *Ceratopteris richardii* [67], which suggest a particular role during female gametophyte development.

Based on the information presented here as well as other reported in the articles commented in this Discussion [37,38,51,52,62] we propose the hypothetical model for *TGS1* action shown in Figure 7, which could be useful to plan future experiments regarding the elucidation of the apomixis molecular control. According to our model, in sexual plants, *TGS1* is expressed in the ovule chalaza and the anther tapetum, where it promotes the generation of mature miR2275 molecules. In turn, miR2275 produces 24nt-meiotic phasiRNAs, which are redirected to the MMC and the PMCs to promote the entrance into meiosis, which occurs concurrently with the acquisition of a gametophytic fate (i.e., the commitment to eventually form a gametophyte) via induction of *QGJ.* Besides, *TGS1* repress the expression of the processed *QGJ* variant within its own expression domain by a still unknown mechanism. In apomictic plants and antisense *tgs1* lines, the expression of *TGS1* is attenuated, thus mature miR2275 are not effectively formed and the delivery of the meiotic 24nt-phasiRNAs becomes compromised. Therefore, meiosis in both the MMC and PMCs is impaired. Moreover, the absence of *TGS1* in the chalaza allows an ectopic expression of *QGJ* at this location and the concomitant induction of a gametophytic fate in nucellar somatic cells, with the consequent emergence of extra non-reduced gametophytes (Figure 7).

After considering the growing amount of evidence reported in the last few years, *TGS1* shapes up as a major promoter of sexuality and inhibitor of aposporous apomixis [38] and emerges as a possible candidate for a reversible switch going from one mode of reproduction to the other. Since *TGS1* is a well-known controller of the splicing and transcription machinery in other organisms, the identification of the molecular routes underlying its function in the plant ovule will certainly lead to a general conception of the mechanisms modulating the reproductive developmental transitions, a requirement for thoughtful ecological evaluation and future harnessing of clonal reproduction into the breeding of major crops.

## 4. Materials and Methods

### 4.1. Plant Materials

The *P. notatum* genotypes used in this work were: Q4117, an apomictic natural tetraploid accession (2n = 4x = 40), originated from Southern Brazil [68]; Q4188: a sexual tetraploid accession (2n = 4x = 40) derived from the cross Q3664 × Q3853 [69]; C4-4x: a sexual tetraploid accession (2n = 4x = 40) derived from chromosome duplication of the diploid plant C4-2x (2n = 2x = 20) after colchicine treatment [26]; three apomictic (JS-40, JS-112, JS-65) and three sexual (JS-83, JS-36, JS-58) full-sib tetraploid (2n = 4x = 40) F_1_ hybrids, originated from the cross of Q4188 × Q4117 [27]; two *tgs1* antisense lines with reduced *TGS1* expression (E2.9: *tgs1-1*; E2.13: *tgs1-2*) [38]; *qgj1* (RNAi-1) and *qgj2* (RNAi-2): two *qgj* RNAi lines with reduced *QGJ* expression [51]; TC1: a transformation control subjected to the same transformation procedure used to obtain the *qgj* RNAi lines, carrying the reporter plasmid pact1-gfbsd2 but not the hairpin construction [51]. All plants are established at IICAR, CONICET-UNR, Rosario, Argentina.

### 4.2. Identification of Splice Variants in Floral Transcriptomes

Splice variants were initially identified from a long-read 454/Roche (Branford, CT, USA) FLX+ (454 Life Sciences Corporation, Branford, Connecticut, USA) reference floral transcriptome of sexual and apomictic *P. notatum*, in which raw data originated from floral samples had been assembled de novo into sexual, apomictic and global (sexual + apomictic) transcriptomes (NCBI Bioproject: PRJNA330955) [34]. DETs identified using the global transcriptome as reference were ordered according to the False Discovery Rate (FDR) value [34]. The top 316 genes with the lowest FDRs were selected to investigate the presence of alternative splice variants in the apomictic (NCBI SRX1971037) and the sexual (NCBI SRX1971038) assemblies. The apomictic and sexual transcripts were aligned with Clustal Omega (http://www.ebi.ac.uk/Tools/msa/clustalo/) (accessed on 1 January 2020). Besides, alignments were made with BLASTN/BLASTX against the NCBI non-redundant databases. The differential expression of particular splice variants was also tested in Illumina HiSeq floral transcriptome libraries [36].

### 4.3. Qualitative PCR 

Qualitative PCR amplifications were performed on cDNA samples generated from floral total RNA. Primers pairs used are shown in Appendix A and were designed with Primer3 v.0.4.0 (http://bioinfo.ut.ee/primer3-0.4.0/) (accessed on 1 January 2020). To validate splice variants, one of the primers was located on internal sequences of the putative intron and the other one on the adjacent exon. Total RNA was extracted from spikelets at stage VII (i.e., with highest *TGS1* expression) according to the *Paspalum* reproductive calendar [33] and using the SV Total RNA Isolation Kit (PROMEGA, Madison, WI USA). cDNAs were synthetized with Superscript II (INVITROGEN, Carlsbad, CA, USA) following the manufacturers’ recommendations. PCR reactions were carried out in a final volume of 25 µL containing: 50 ng cDNA, 0.5 µM of each specific primer (Table 1), 0.2 mM dNTPs, 1 × PCR buffer (Promega), 2.5 mM MgCl_2_ and 1.25 µL of Taq Polymerase (Promega). Cycling was completed in a MyCycler thermocycler (BIORAD, Hércules, CA, USA), as follows: 2 min at 95 °C, 35 cycles of 1 min at 95 °C, 1 min at 57 °C and 1 min at 72 °C, and a final elongation of 5 min at 72 °C. Amplified fragments were visualized onto 2% agarose gels, electrophoresed at 60 mA in TAE 1× buffer and stained with 10 mg/L ethidium bromide.

### 4.4. qPCR Experiments

For qPCR, total RNA was extracted from spikelets at the required developmental stage [33] by using the SV Total RNA Isolation Kit (PROMEGA, Madison, WI, USA). Primers pairs used are shown in Appendix A and were designed with Primer3 v.0.4.0 (http://bioinfo.ut.ee/primer3-0.4.0/) (accessed on 1 January 2020). cDNAs were synthetized with Superscript II (INVITROGEN, Carlsbad, CA, USA) following the manufacturers’ recommendations. Quantitative PCR reactions (final volume: 20 µL), were performed in a Rotor-Gene Q thermocycler (QIAGEN, Hilden, Germany) and included: 0.5 µM gene-specific primers (Appendix A), 1× Real Mix qPCR (BIODYNAMICS, Buenos Aires, Argentina) and 20 ng of cDNA. In each experiment, two biological replicates were processed, in which each determination was run in three technical replicates. The constitutive gene β-TUBULIN was used as housekeeping reference, since in previous work this gene had been selected as the best reference for comparisons in this particular floral apomictic-sexual *Paspalum* system [70,71,72]. Negative controls were processed without template DNA. Amplifications were performed with the following program: 2 min at 94 °C, 45 cycles of 94 °C for 15 min, 57 °C for 30 s, and 72 °C for 17 s, and a final elongation step of 5 min at 72 °C. Relative quantitative expression levels were assessed using the REST-RG 2009 software (QIAGEN, Hilden, Germany).

### 4.5. Stem-Loop qPCR Experiments

Total RNA was extracted from flowers of plants E2.9 (*tgs1-1*), C4-4x (sexual control) and Q4117 (apomictic control) at meiosis, using the SV Total RNA Isolation Kit, following the *Paspalum* calendar [33] as a guide. miRNA specific stem loop primers (Appendix A) were designed using sRNAPrimerDB (http://www.srnaprimerdb.com) (accessed on 1 January 2020) according to the protocol described by Xie et al. (2018) [73]. For reverse transcription, we used the ImProm-II™ Reverse Transcription System, following the manufacturers’ recommendations. Each PCR reaction (20 μL) contained: 1X Real mix (Biodynamics, Buenos Aires, Argentina), primers upper and lower 0.5 µM, cDNA 20 ng and sterile distilled water to complete volume. Cycling was completed as follows: hold at 94 °C, 2 min; cycling (45×) at 94 °C-15″, 57 °C-30″, 72 °C-20″. Negative controls without template were included in all experiments. 

### 4.6. In Situ Hybridization

Spikelets of genotypes *tgs1-1* (E2.9) and Q4188 (sexual control) were collected at premeiosis/meiosis, fixed in a solution of 4% paraformaldehyde/0.25% glutaraldehyde/0.01 M phosphate buffer (pH 7.2). Sixty (60) ovaries per plant were dehydrated in an ethanol/xylol series and embedded in paraffin. Specimens were sliced into 10-μm-thin sections and placed onto slides treated with 100 μg mL^−1^ poly-L-lysine. Paraffin was removed with a xylol/ethanol series. Both sense (T7) and antisense (SP6) RNA probes were produced using a plasmid containing the QGJ clone [51]. Probes were labelled using a Roche DIG RNA labelling kit (Indianapolis, IN, USA) (SP6/T7) (ROCHE DIAGNOSTIC CORPORATION PO, INDIANAPOLIS, IN, USA) and hydrolyzed into 150–200-bp fragments. Following prehybridization at 37 °C for 10 min in 0.05 M Tris–HCl buffer (pH 7.5) containing 1 μg mL^−1^ proteinase K, hybridization at 37 °C was performed overnight in a buffer containing 10 mM Tris–HCl (pH 7.5), 300 mM NaCl, 50% deionized formamide, 1 mM EDTA (pH 8), 1× Denhardt’s solution, 10% dextran sulphate, 600 ng mL^−1^ total RNA, and 60 ng of the corresponding probe. Detection was performed following the Roche DIG detection kit instructions using anti-DIG AP and NBT/BCIP as substrates.

## 5. Conclusions

The RNA methyltransferase *TGS1* participates in the alternative splicing of a *LHC Ib-21*, a protein of the chloroplastic light harvesting antenna with a predicted uncharacterized moonlighting function related with RNA regulation. However, *TGS1* is responsible for the expression level of a processed transcript encoding the mitogen activated protein kinase kinase kinase (MAP3K) QGJ in florets of sexual plants. Moreover, in sexual *Paspalum* flowers, *TGS1* induces the expression of miR2275, a miRNA responsible for the synthesis of essential meiotic 24-nt phasi-RNA previously characterized in maize and rice. In the context of previous results, the evidence shown here suggests that the *TGS1* action on *QGJ* and miR2275 might: (i) trigger a molecular route leading to the acquisition of gametophyte commitment and the induction of meiosis in the MMC and pollen mother cells, and (ii) repress gametophyte commitment in cells surrounding the MMC, so in absence of *TGS1*, the spatial *QGJ* distribution is altered, inducing the formation of unreduced supernumerary gametophytes. 

## Figures and Tables

**Figure 1 plants-11-01929-f001:**
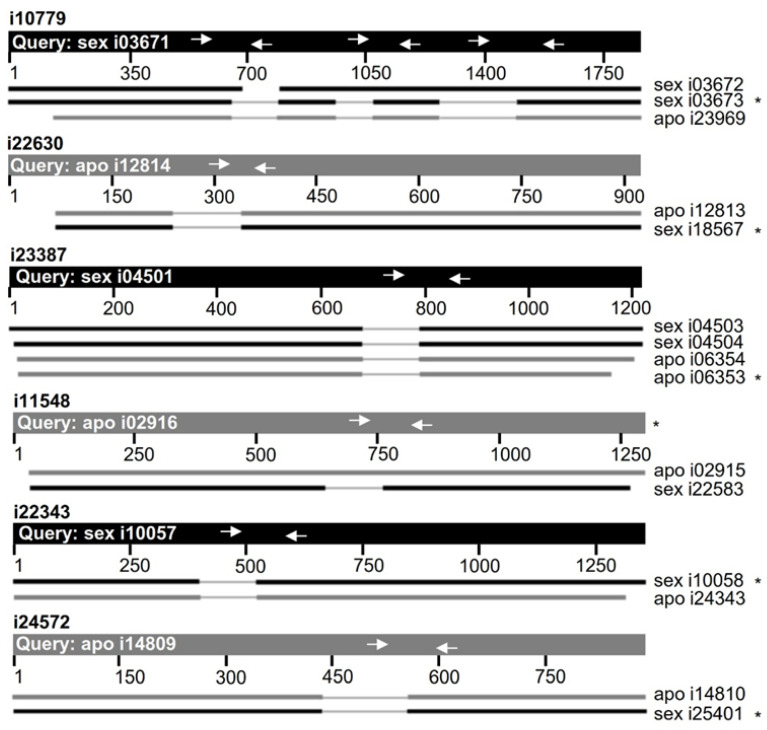
Representation of particular splice variants in florets of sexual and apomictic *P. notatum* plants. Structural scheme of DETs i10779, i23387, i11548, i22343, i24572 and i22630 variants expressed in apomictic and sexual 454/Roche FLX+ libraries [34]. We used the NCBI BLASTN alignment tool to compare all homologs. The query applied to produce the alignments was denoted at the top as a thicker line and corresponded always to an unprocessed variant (with intron). Arrows inside the query mark the position of the primers designed for the qPCR assays. Since one of the primers was always located within the putative intron, the primer pair was capable of amplifying only the unprocessed variant. Sexual isotigs are represented in black and apomictic ones in gray. The variant detected as differential in the 454/Roche FLX+ libraries was marked with an asterisk (*).

**Figure 2 plants-11-01929-f002:**
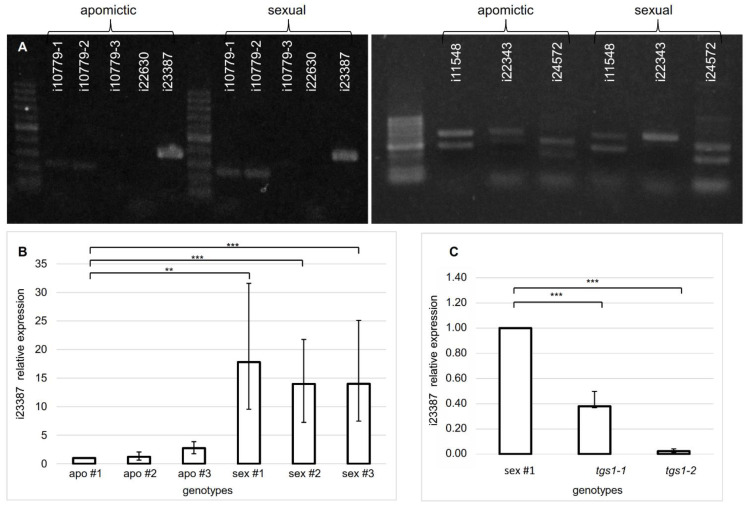
Qualitative and quantitative amplification of unprocessed (+intron) transcript variants. (**A**) PCR amplification of transcripts with putative differential splice variants from cDNAs of apomictic (Q4117) and sexual (C4-4x) plants. i10779-1/2/3 correspond to three introns present in this transcript. Molecular weight marker: 50–500 pb. (**B**) Relative expression quantification of the i23387 unprocessed variant in florets of apomictic (JS65, JS112, JS40) and sexual (JS83, JS36, JS58) F_1_ genotypes. (**C**) Relative expression quantification of the i23387 unprocessed variant in flowers of a sexual control plant (Q4188) and two antisense *tgs1* lines (*tgs1-1*: E2.9; *tgs1-2*: E2.13). Asterisks denote the significance (** *p* < 0.01, *** *p* < 0.001).

**Figure 3 plants-11-01929-f003:**
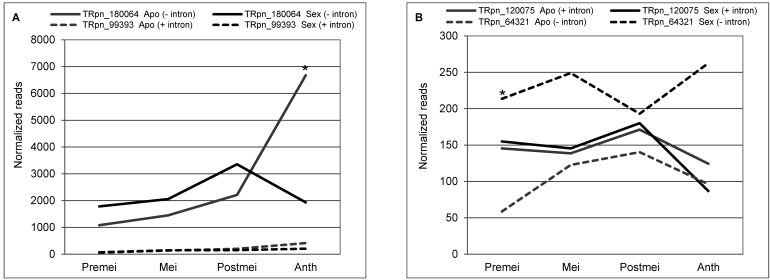
Expression dynamics of the splice variants of *LHC Ib-21* and *QGJ* along sexual and apomictic development. (**A**) Normalized read counts corresponding to the processed (TRpn_180064) and non-processed (TRpn_99393) *LHC Ib-21* splice variants. Note that the processed splice variant is expressed at higher levels than the non-processed and it is differentially represented in apomictic and sexual plants at anthesis. No significant differential expression could be detected for the non-processed variant in RNAseq experiments, although in qPCR analysis (known to be more sensitive than sequencing approaches) the non-processed variant is consistently detected overexpressed in sexual plants at anthesis. (**B**) Normalized read counts corresponding to the processed (TRpn_64321) and non-processed (TRpn_120075) *QGJ* splice variants. The processed splice variant is upregulated in sexual plants at premeiosis and remains with higher expression along development (yet not at statistically significant levels). The non-processed variant is expressed at low levels. Premei, Mei, Postmei and Anth indicate ovules at the developmental stages premeiosis, meiosis, postmeiosis and anthesis, respectively. The asterisk denotes the differential expression significance (* *p* < 0.05).

**Figure 4 plants-11-01929-f004:**
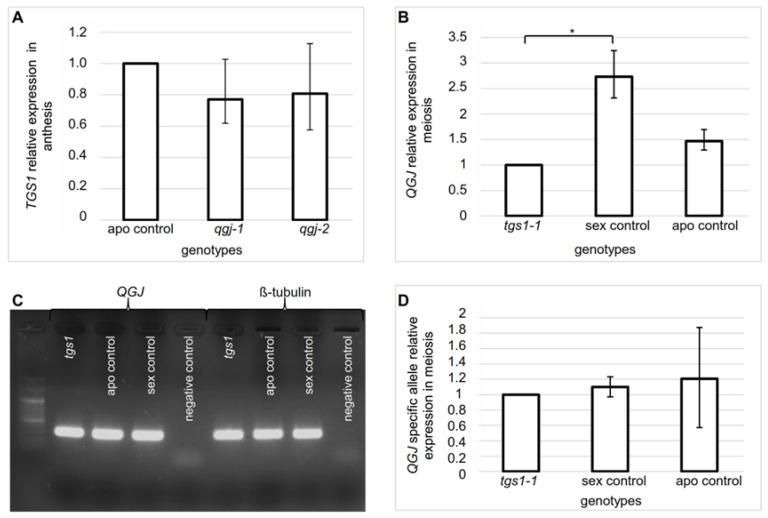
Analysis of a mutual regulatory effect involving *TGS1* (apospory repressor) and *QGJ* (apospory inducer). (**A**) Expression of *TGS1* in two *qgj* lines (*qgj1*: RNAi1; *qgj2*: RNAi2) and an apomictic control (TC1) at anthesis, the stage of maximum *TGS1* expression. (**B**) Expression of *QGJ* in a *tgs1* genotype (*tgs1-1*: E2.9), a sexual control (C4-4x) and an apomictic control (Q4117) at meiosis. (**C**) Amplification products of the non-processed *QGJ* form and the housekeeping gene β-*tubulin* after qPCR assay (after 35 cycles of amplification). (**D**) Expression of the *QGJ* unprocessed variant (QGJ + intron) in a *tgs1* genotype (*tgs1-1*: E2.9), a sexual control (C4-4x) and an apomictic control (Q4117), at meiosis. Asterisks denote the significance at *p* < 0.05 (*).

**Figure 5 plants-11-01929-f005:**
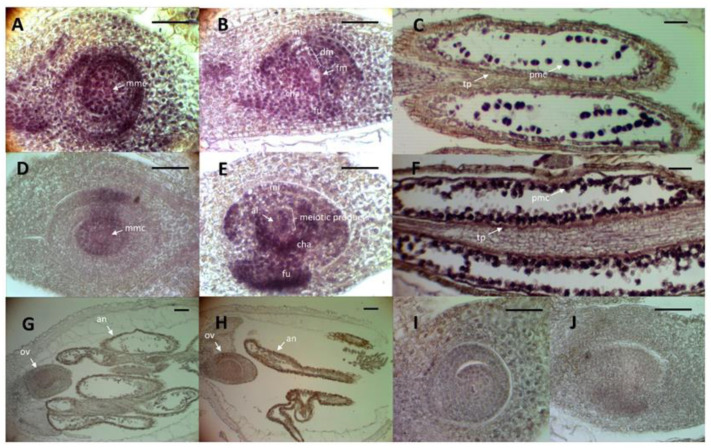
In situ hybridization of *QGJ* in *tgs1* lines and controls. (**A**–**F**) Hybridization with the antisense probe. (**G**–**J**) Hybridization with the sense probe. (**A**) Premeiotic ovule of Q4188 (sexual plant): the signal is observed in the whole ovule, including the MMC. (**B**) Postmeiotic ovule of Q4188; the signal was transferred to the micropylar region and the funiculus; the functional megaspore has no signal; the degenerating megaspores are with signal; (**C**) Anthers of Q4188: no signal is observed in the tapetum; almost all pollen mother cells display signal (**D**) Premeiotic ovule of *tgs1.1*: moderate to low signal is observed in the whole ovule, including the MMC (**E**) Postmeiotic ovule of *tgs1.1*: the signal was transferred to the chalaza and the funiculus and is detected inside all meiotic products, which are surrounded by a cell layer with lower signal; an enlarged cell reminiscent to an apospory initial is located aside the meiotic products, within the less-hybridized area; (**F**) Anthers of *tgs1.1*: moderate signal is observed in the tapetum; only some of the pollen mother cells display strong signal, the rest remains unstained. (**G**–**H**) meiotic ovules of sexual and *tgs1* lines, respectively, hybridized with *QGJ* sense probe (10×). (**I**–**J**) premeiotic ovules of sexual and *tgs1* lines, respectively, hybridized with *QGJ* sense probe (40×). ai: a putative apospory initial cell; an: anthers; cha: chalaza; dm: degenerated megaspores; fm: functional megaspore; fu: funiculus; mi: micropyle; mmc: megaspore mother cell; pmc: pollen mother cells; tp: tapetum. Bars: 20 μm.

**Figure 6 plants-11-01929-f006:**
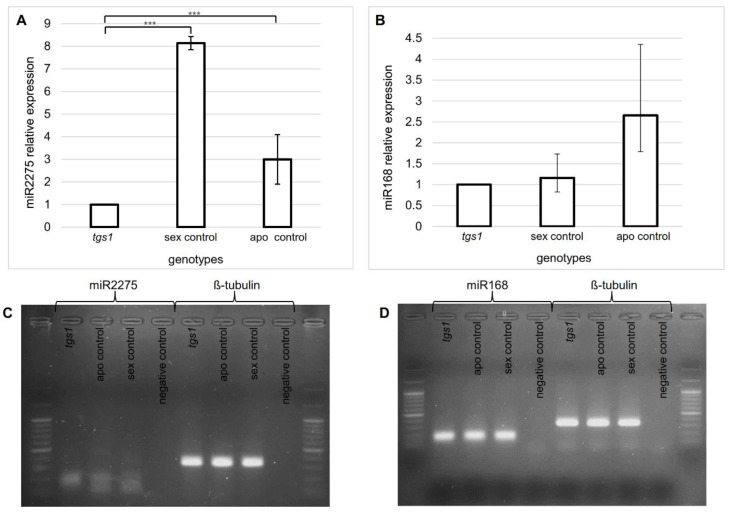
Quantitative stem-loop PCR of miR2275 and miR168. (**A**) miR2275 relative expression in the *tgs1-1* antisense line, a sexual control (sexual C4-4x) and an apomictic control (Q4117). (**B**) miR168 relative expression in *tgs1-1* antisense lines, a sexual control (C4-4x) and an apomictic control (Q4117). (**C**,**D**) Final products of the stem loop amplification for miR2275 and miR168, respectively (after 35 cycles of amplification). Asterisks denote the significance at *p* < 0.001 (***).

**Figure 7 plants-11-01929-f007:**
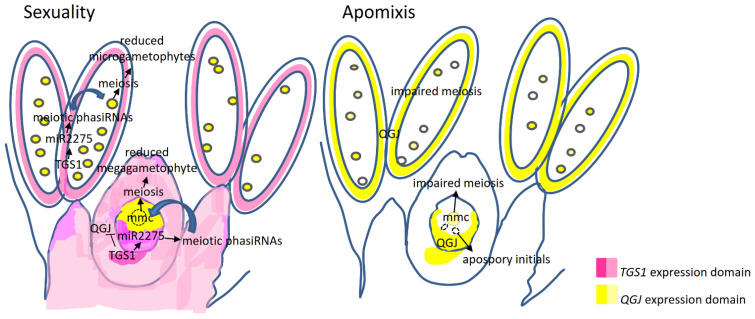
Hypothetical model of *TGS1/QGJ* functional interactions in the ovule. *TGS1* could be repressing the expression of *QGJ* within its own activity domain, but inducing it in the germinal line via a non-cell autonomous miR2275/24-phasiRNA-mediated mechanism. In absence of *TGS1* (i.e., apomictic genotypes and *tgs1* antisense lines) *QGJ* would overexpress ectopically across the canonical *TGS1* activity domain (the anther tapetum and the chalaza). Around the MMC, a limited number of cell layers show low QGJ activity, allowing the differentiation of apospory initials (i.e., cells with functional megaspore identity, but non-reduced). Besides, the absence of *TGS1* is also associated with a decrease in miR2275 and the loss of the miR2275/24-phasiRNA pathway. The latter condition compromises the induction of *QGJ* within MMCs and pollen mother cells, impairing proper differentiation of the meiotic germline. mmc: megaspore mother cell.

**Table 1 plants-11-01929-t001:** Transcripts presenting putative splice variants with differential representation in apomictic and sexual floral transcriptomes of *P. notatum*.

Transcript (Global Assembly) *	Reads Apo *	Reads Sex *	Annotation	*p*-Value for Differential Expression	FDR for Differential Expression
i10779	139	458	Tetraketide alpha-pyrone reductase 1	8.99 × 10^−38^	1.28 × 10^−34^
i22630	144	377	Strictosidine synthase	2.21 × 10^−23^	1.18 × 10^−20^
i23387	490	235	Chlorophyll a-b binding protein 1B-21	2.98 × 10^−22^	1.50 × 10^−19^
i11548	729	411	Chlorophyll a-b binding protein CP26	8.11 × 10^−22^	3.99 × 10^−19^
i22343	164	397	LTP_2 Probable lipid transfer	1.84 × 10^−21^	8.95 × 10^−19^
i24572	24	105	Dehydrin DHN1	8.14 × 10^−16^	2.42 × 10^−13^

* According to the 454/Roche FLX+ assemblies reported in Ortiz et al. (2017) [34].

## Data Availability

The RNAseq sequence datasets used in this work are openly available at the NCBI repository under the accession numbers SRX1971037 y SRX1971038 (454/Roche FLX+), SRP099144 (small RNAs) and PRJNA511813 (Illumina sequencing). All plant materials used here are maintained at the IICAR greenhouses or controlled chambers and are available after official request for academic purposes.

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
