# Peer review of "Spotting the Targets of the Apospory Controller TGS1 in Paspalum notatum"

_plants, 2022, doi:10.3390/plants11151929_

Round 1

Reviewer 1 Report

A highly technical research report that could be used for other species. Not detracting from the work, if simplified (but not suggested) would be of value for those less experienced in the topic of apomixis. Regarding the species studied, if the research findings are extended at the field level, it could benefit significantly regions that have poor soils eg Australia, USA, Southern Africa and America as well as the Indo-Asia and this should be noted. Regarding food value, this should be clarified, that it is a forage crop for animals but equally of benefit to soil conservation. The practical value of the research could be challenged in regions where soil quality is high as Paspalum notatum is highly competitive hence the risk of displacing other useful forage species. Furthermore, there is no mention of the agronomics of the novel offsprings, the nutritive value of the crop and its palatability to animals.  All of these items should be introduced or discussed in the appropriate parts of the research report.

Author Response

We would like to thank Reviewer 1 for his/her valuable comments, which will be helpful to improve the manuscript. Please find our answers below:

Reviewer 1 comment 1: Not detracting from the work, if simplified (but not suggested) would be of value for those less experienced in the topic of apomixis.

Our answer to Reviewer 1 comment 1:

We tried to simplify the structure of some sentences across the manuscript, or directly eliminate them, in order to make the text more accessible to those readers less experienced in the topic of apomixis (we agree with Reviewer 1 on the complexity of the field). The following changes were included:

Page 2, line 79

“Population studies revealed that the genetic variation generated in diploids may be pumped up to polyploids (which are capable of reproduce clonally) through the formation of triploid bridges.”

was changed to

“Population studies revealed that sexual diploids can occasionally form new apomictic polyploids via hybridization of non-reduced gametes. Therefore, the genetic variability present in sexual diploid populations can be pumped up to apomictic polyploid ones.”

Page 3, line 119:

“To start with the identification of molecular pathways modulated by TGS1 in P. notatum ovules, we evaluated putative targets showing differential processing in an attenuated expression context. First, we mined the available P. notatum sexual and apomictic reproductive transcriptomes in search for differentially represented splice variants and miRNAs, building on the contrasting TGS1 activity naturally shown by these reproductive biotypes.”

was changed to

“To initiate the identification of molecular pathways modulated by TGS1 in P. notatum ovules, we decided to evaluate a group of RNA molecules showing differential processing in plants with attenuated TGS1 expression. First, we compared sexual and apomictic floral transcriptomes in search for differentially represented mRNA splice variants and miRNAs, taking advantage of the contrasting TGS1 activity previously described for these reproductive biotypes [37]”.

Pag 3, line 124

“Then, we carried out intron-specific (for mRNAs) or stem-loop (for miRNAs) qPCR analysis in sexual tgs1 antisense lines as well as in apomictic and sexual controls”

was changed to

“Then, we carried out intron-specific (for mRNAs) or stem-loop (for miRNAs) qPCR analysis in sexual tgs1 antisense lines as well as in apomictic and sexual controls, to confirm a causal association between the TGS1 knock-down and the emergence of particular splice variants/miRNAs.”

In pag. 6, line 205, to segment the text and make it easier to read, we created a new paragraph and eliminated the word “Moreover”.

“…which was described as necessary for the formation of AESs in Paspalum

was changed to

“…which is necessary for the formation of AESs in Paspalum

In page 10, line 347, the sentence

“However, since TGS1 is naturally upregulated in spikelets of sexual plants, we decided to investigate them in search of overrepresented miRNAs.”

Was changed to

“However, since TGS1 is naturally upregulated in spikelets of sexual plants, we decided to search miRNAs which were overrepresented in floral transcriptome libraries of sexual plants with respect to apomictic ones.”

Pag 14, line 512

The sentence “While QGJ expression can still be detected in the MMC and the meiocyte, it is detected at lower levels in PMC, possibly compromising the formation of the gametophyte” was eliminated.

Reviewer 1 comment 2: Regarding the species studied, if the research findings are extended at the field level, it could benefit significantly regions that have poor soils eg Australia, USA, Southern Africa and America as well as the Indo-Asia and this should be noted. Regarding food value, this should be clarified, that it is a forage crop for animals but equally of benefit to soil conservation.

Our answer to Reviewer 1 comment 2:

In pag.2, line 77, we added the following sentence:

“Moreover, it is recognized as a valuable forage grass, which also provides benefits in terms of soil conservation [20-21].”

Reviewer 1 comment 3: The practical value of the research could be challenged in regions where soil quality is high as Paspalum notatum is highly competitive hence the risk of displacing other useful forage species. Furthermore, there is no mention of the agronomics of the novel offsprings, the nutritive value of the crop and its palatability to animals.  All of these items should be introduced or discussed in the appropriate parts of the research report.

Our answer to Reviewer 1 comment 3:

The objective of our research was not particularly focused in the Paspalum notatum breeding and/or their introduction as an exotic crop. We are trying to characterize the molecular control of aposporous apomixis in a Gramineae model in order to be eventually able to transfer this knowledge to the breeding of sexual crops like rice or maize. However, as mentioned by Reviewer 1, the characterization of the molecular basis of apomixis could also contribute to the apomixis-based Paspalum breeding programs that are currently conducted at IBONE-CONICET Argentina. Regarding this point, in the new version of the manuscript we introduced the following changes in the Discussion section:

Pag 11, line 389, the first two paragraphs of the Discussion section were modified as follows, to include information and references on the Paspalum notatum breeding:

“In the last few years, an unprecedented amount of data originated from genome and transcriptome sequencing projects have flooded the apomixis field. Hundreds of candidate genes showing differential expression in sexual and apomictic plants were identified at different reproductive developmental stages [53]. However, little is known on the functional role of these candidates, the operative interactions among them or the identity of the molecules controlling the coexistence of a subtle balance between both reproductive types in natural populations. The identification of molecular markers co-segregating with apomixis, the generation of artificial sexual polyploids after colchicine duplication, the construction of transcriptome and genomic databases and the establishment of biolistic transformation platforms represent good perspectives for selecting traits of interest in natural apomictic species like P. notatum [21]. In fact, heterosis for forage yield and cold tolerance has been repeatedly reported in this species and an upright and fast-growing apomictic P. notatum hybrid was recently released as a forage cultivar [21]. Instead, the harnessing of apomixis in major crops like rice and maize requires a much more detailed knowledge of the molecular mechanisms controlling the balance between apomixis and sexuality. The establishment of such programs requires sophisticated molecular tools and might bring new ecological challenges related with spreading the trait via pollen and seeds (e.g., uniparental reproduction, unidirectional gene transfer) [17]. Therefore, before any attempt of using the trait, we should expand our information on the molecular, functional and organizational mechanisms operative in natural apomictic plant populations, to improve our capacity to evaluate and avoid any damage.

Natural apomicts display a wide range of developmental approaches to balance their competence for both genetic variation and cloning. Both features can coexist in the same species (i.e., confined at different ploidy levels), within the same plant (i.e., in facultative apomicts) and even the same ovule (i.e., occurrence of polyembryony of sexual and apomictic origin). Moreover, the proportion of offspring formed by each reproductive mode can be influenced by the environment [17]. In any case, there seem to be regulators operating to favor apomixis or sexual reproduction in different contexts (variable ploidy, diverse genetic backgrounds, particular environmental conditions), which could be harnessed to abolish the ecological impact of the trait ensuring a safe use in agriculture. Understanding the functional interactive dynamics of both reproductive modes at the molecular level will be crucial to predict how an apomictic crop may behave in natural fields and visualize potential ecological threats.”

Finally, we carefully examined the whole manuscript and corrected several typos and grammar errors (please, control the tracked copy).

.

Thank you very much for your valuable corrections.

Reviewer 2 Report

The basic understanding of the paper summarizes, Spotting the targets of the apospory controller TGS1 in Paspa-2 lum notatum.

Authors mined the available P. notatum sexual and apomictic reproductive transcriptomes in search for differentially represented splice variants and miRNAs, building on the contrasting TGS1 activity naturally shown by these reproductive biotypes. Then, carried out intron-specific (for mRNAs) or stem-loop (for miR-124 NAs) qPCR analysis in sexual tgs1 antisense lines as well as in apomictic and sexual controls. Identified three transcripts whose processing or expression 126 is altered in tgs1 antisense lines, i.e., LHC Ib-21 (encoding CHLOROPHYLL A-B BINDING 127 PROTEIN 1B-21. MS results provided information on TGS1 functional targets.

Some spell checks are required.

Author Response

Reviewer 2 comment 1:

Authors mined the available P. notatum sexual and apomictic reproductive transcriptomes in search for differentially represented splice variants and miRNAs, building on the contrasting TGS1 activity naturally shown by these reproductive biotypes. Then, carried out intron-specific (for mRNAs) or stem-loop (for miR-124 NAs) qPCR analysis in sexual tgs1 antisense lines as well as in apomictic and sexual controls. Identified three transcripts whose processing or expression is altered in tgs1 antisense lines, i.e.LHC Ib-21 (encoding CHLOROPHYLL A-B BINDING PROTEIN 1B-21. MS results provided information on TGS1 functional targets.

Our answer to Reviewer 2 comment 1:

Thank you for your evaluation of the work regarding the identification of TGS1 targets.

Reviewer 2 comment 2: Some spell checks are required.

Our answer to Reviewer 2 comment 2:

We carefully examined the whole manuscript and corrected several typos and grammar errors (please control the tracked copy).

Thank you very much for your valuable corrections.
